

# Unveiling the role of gasdermin B in cancer and inflammatory disease: from molecular mechanisms to therapeutic strategies

Weixiao Yang[1,*], Xu Hu[1,*] and Xiang Li[1]

West China Medical School, Sichuan University, Department of Urology, Institute of Urology, Chengdu, Sichuan, China

[*] These authors contributed equally to this work.

## ABSTRACT

Gasdermin B (GSDMB) is a member of the gasdermin (GSDM) protein family, primarily known for mediating pyroptosis, an inflammatory form of programmed cell death. Recent studies have revealed the diverse molecular functions of GSDMB and its close association with various diseases, particularly cancers (*e.g.*, breast cancer, gastric cancer, bladder cancer) and inflammatory diseases (*e.g.*, asthma, inflammatory bowel disease). At the molecular level, GSDMB induces pyroptosis by forming pores in the cell membrane, leading to membrane rupture. This function is common across the GSDM protein family; however, GSDMB also exhibits unique non-pyroptotic functions, such as modulating cell proliferation, migration, and immune responses. In multiple cancers, including breast cancer, gastric cancer, and cervical cancer, high expression of GSDMB correlates with poor prognosis, promoting cancer cell proliferation, invasion, and metastasis through interactions with signaling pathways such as STAT3 and mitogen-activated protein kinase (MAPK)/extracellular signal-regulated kinase (ERK). Additionally, GSDMB influences the immune microenvironment through its pyroptotic activity, playing a role in the initiation and regulation of inflammation. Upon activation, it can directly cleave target cells *via* its N-terminal domain, contributing significantly to chronic inflammatory diseases and NK cell-mediated antibacterial responses. In conclusion, as a multifunctional protein, GSDMB not only participates in pyroptosis but also regulates non-pyroptotic processes, playing an important role in cancer progression and inflammatory diseases. Further elucidating the detailed mechanisms of GSDMB may offer novel therapeutic avenues for these conditions.

## INTRODUCTION

The human gasdermin (GSDM) superfamily comprises GSDMA/B/C/D, GSDME (also known as DFNA5), and DFNB59 (Pejvakin, PJVK), with corresponding homologs in mice (GSDMA1-3, GSDMC1-4, GSDMD, DFNA5, and DFNB59) (*Bergsbaken et al., 2011*; *Ding et al., 2016*; *Tamura et al., 2007*). The GSDM family is known to regulate cellular

Corresponding author
Xiang Li, xiangli87@hotmail.com

proliferation, differentiation, and programmed cell death, especially pyroptosis. Pyroptosis, first defined in 2015, is GSDM-mediated programmed cell death (*Shi et al., 2015*).

The *GSDM* gene family was first identified in the early 21st century as candidate genes for alopecia-like skin mutations in mice (*Sato et al., 1998*). The name "GSDM" originates from the expression of GSDM in the gastrointestinal tract and skin. GSDM proteins exhibit selective expression across various mucosal tissues, especially during infection (*Liu & Lieberman, 2020*; *Saeki et al., 2000*). For instance, GSDMA is found in the skin and gastrointestinal tract; GSDMB in the lungs, esophagus, gastrointestinal tract, and immune cells; GSDMC in keratinocytes and the gastrointestinal tract; and GSDMD in the gastrointestinal epithelium, macrophages, and dendritic cells of the immune system (*Liu & Lieberman, 2020*). GSDME is uniquely expressed in mesenchymal cells, such as skeletal muscle, myocardium, the central nervous system, and placenta (*Van Laer et al., 1998*).

Early studies also pointed out that the N-terminal domain of GSDM shares significant sequence similarity with DFNA5 (Dominant Familial Nonsyndromic Hearing Loss, locus 5) (*Van Laer et al., 1998*). Based on this homology, several other members of the GSDM family have been identified. Notably, *GSDMB* is the only *GSDM* gene that has not been found in rodents (*Das et al., 2016*). This observation suggests that *GSDMB* is not a general component of the mammalian immune system, but rather has evolved a unique function in humans and other mammals (*Hansen et al., 2021b*). In the GSDM family, research focusing on *GSDMD* and *GSDME* is relatively abundant, and studies on the functions and mechanisms of *GSDMB* are progressively deepening. Since 2007, *Moffatt et al. (2010)* and *Moffatt et al. (2007)* have conducted large-scale GWAS studies, revealing that single nucleotide polymorphisms (SNPs) in *GSDMB*, particularly rs7216389, are strongly associated with asthma susceptibility. Located in the intronic region of the GSDMB gene, this SNP regulates the expression of ORMDL3, playing a significant role in childhood asthma development. Subsequent GWAS studies have also revealed associations between *GSDMB* and various inflammatory and immune-related diseases, including inflammatory bowel disease (IBD), chronic rhinosinusitis, primary biliary cirrhosis, and cervical cancer, among others (Table 1). Although *GSDMB* has been implicated in numerous GWAS studies and associated with various diseases, the precise mechanisms by which it contributes to these conditions remain insufficiently explored. There is a need for further investigation and verification of its molecular functions and role in disease pathology, which is crucial for translating these findings into therapeutic strategies (*Chao, Kulakova & Herzberg, 2017*; *Ding et al., 2016*; *Handunnetthi et al., 2021*; *Hitomi et al., 2017*; *Lutkowska et al., 2017*; *Zack et al., 2021*).

In this review we provide a comprehensive summary based on the molecular characteristics and diverse functions of GSDMB. It details the roles of GSDMB in cancers, autoimmune and inflammatory diseases, as well as bacterial/viral infectious diseases, as evidenced by existing research, and explores its potential as a prognostic marker for these diseases. Furthermore, we summarize the latest research on GSDMB as a therapeutic target, offering new directions for the development of future strategies targeting GSDMB. By integrating these aspects, our review presents a broader and deeper perspective, providing fresh insights for future research.

**Table 1  The association between *GSDMB* SNPs variants and diseases.**

| Disease | SNPs | Potential functional impact | References |
|---|---|---|---|
| Asthma | rs7216389, rs1031458, rs3902920, rs11078928 | Regulates *ORMDL3* expression, influences airway inflammation and remodeling | *Li et al. (2021)* and *Morrison et al. (2013)* |
| Inflammatory Bowel Disease (IBD) | rs2872507 | Downregulates *GSDMB* expression, affects epithelial restitution | *Söderman, Berglind & Almer (2015)* |
| Chronic Rhinosinusitis | rs7216389 | Associated with airway inflammation and remodeling | *Zack et al. (2021)* |
| Type 1 Diabetes (T1D) | rs12150079, rs2305480, rs3894194, rs12936231 | Regulates *ORMDL3* and *ZPBP2* expression, impacts immune response | *Barrett et al. (2009)*, *Verlaan et al. (2009a)* and *Witsø et al. (2015)* |
| Primary Biliary Cholangitis | rs12946510 | Influences *ORMDL3* and *GSDMB* expression; linked to autoimmunity | *Hitomi et al. (2017)* |
| Cervical Cancer | rs8067378 | Increases *GSDMB* expression, linked to cancer progression | *Lutkowska et al. (2017)* |

## The intended audience and need for this review

This review is designed for researchers and clinicians in molecular biology, oncology, and immunology, with a focus on GSDMB. GSDMB's importance lies in its multifaceted roles in cancer biology and inflammatory diseases. By influencing key processes such as pyroptosis, cell proliferation, and immune modulation, GSDMB significantly impacts cancer progression, metastasis, and inflammatory pathologies. Despite growing research interest, its unique functions and regulatory pathways remain insufficiently explored. This review consolidates current insights into GSDMB's structure, functions, and clinical relevance, offering a valuable resource to advance research and therapeutic development, particularly as GSDMB emerges as a promising target in cancer and immune disorders.

## SURVEY METHODOLOGY

We conducted a comprehensive literature search across PubMed, Google Scholar, and Web of Science to identify studies relevant to this review. The search utilized keywords and Medical Subject Heading (MeSH) terms, including GSDMB, GSDML, Gasdermin B, cancer, neoplasms, hypersensitivity, inflammation, infectious diseases, molecular targeted therapy, prognosis, biomarkers, autoimmune diseases, and pyroptosis, tumor microenvironment, caspases, along with their variants. These terms were systematically sorted, combined, and used in structured database queries. Relevant studies were selected based on a thorough review of their abstracts. Ultimately, 90 articles published between 1998 and 2024 were included in this review.

## Characteristics of the GSDMB gene and protein structure

GSDMB was previously known as GSDML (gasdermin-like protein). The GSDMB gene is located on chromosome 17q21.1 and comprises 11 exons. The 17q21 region may also contain genes affecting diseases associated with abnormal immune responses, such
as asthma, allergic rhinitis, and IBD. Such as *ORMDL3*, which can regulate *GSDMB* expression. The GSDMB protein consists of 411 amino acids The GSDMB protein has four splice variants of different lengths, with GSDMB3 being the longest isoform (416 amino acids) and GSDMB2 the shortest (394 amino acids) (*Das, Miller & Broide, 2017*; *Oltra et al., 2023*). The cis-regulatory elements of *GSDMB* contain two distinct promoters: a cellular promoter, which is expressed exclusively in normal gastric tissue and certain cancer cells, and a long terminal repeat-derived (LTR-derived) promoter, which has been detected in nearly all cancer types and multiple normal tissues. This suggests that the LTR-derived promoter may serve as the primary driver of *GSDMB* expression across various tissues and may facilitate its upregulation in cancer cells. Although existing studies indicate the dominant role of the LTR-derived promoter in *GSDMB* gene expression, the precise molecular mechanisms underlying how the LTR-derived promoter competes with the cellular promoter to regulate *GSDMB* expression, as well as which specific signaling pathways or cellular environmental factors activate the LTR-derived promoter, remain largely unexplored (*Feng, Fox & Man, 2018*; *Komiyama et al., 2010*; *Sin et al., 2006*).

As an "executor" of pyroptosis, all GSDM family members, including GSDMB, except Pejvakin, contain two conserved domains: an N-terminal pore-forming domain and a C-terminal repressive domain. The N-terminal domain of most GSDMs can induce pyroptosis, although this function has not yet been detected in Pejvakin (*Rogers et al., 2017*).

*Oltra et al. (2023)* reported that the *GSDMB* gene (NCBI Gene ID: 55876) produces at least six transcript variants, which are translated into four distinct protein isoforms (GSDMB1-4). These isoforms exhibit differential expression patterns across various diseases, influenced by genetic factors such as single nucleotide polymorphisms (SNPs). For instance, studies by *Panganiban et al. (2018)* and *Morrison et al. (2013)* have demonstrated that the rs11078928 variant impacts the alternative splicing of *GSDMB*, resulting in the loss of exon 6. This exon loss may diminish the pyroptotic activity of GSDMB, thereby potentially reducing the susceptibility to asthma (*Chao, Kulakova & Herzberg, 2017*; *Morrison et al., 2013*). Moreover, *Lutkowska et al. (2017)* identified that the G allele of the rs8067378 SNP may modulate the transcription of *GSDMB*, influencing the proliferation of breast cancer cells. The primary difference among these four translated isoforms lies in the alternative splicing of exons 6 and 7. Exon 6 encodes 13 amino acids, while exon 7 encodes nine amino acids, both of which are located within the flexible interdomain linker region between the N-terminal (NT) and C-terminal (CT) domains. Specifically, GSDMB1 lacks exon 6 (Δ6), GSDMB2 lacks both exons 6 and 7 (Δ6-7), GSDMB3 contains both exons, and GSDMB4 lacks exon 7 (Δ7) (*Oltra et al., 2023*).

## Molecular functions of the GSDMB protein

GSDMB exerts diverse molecular functions through both pyroptosis-dependent and non-pyroptosis-dependent mechanisms, playing distinct roles in various diseases (Fig. 1). Generally, GSDM proteins maintain oligomerization through interactions between their N-terminal and C-terminal domains (*Kuang et al., 2017*). When stimulated by various exogenous or endogenous factors, GSDM proteins are cleaved by certain caspases or

granzymes, releasing the pore-forming N-terminal domain. The N-terminal domain of GSDMs binds to acidic phospholipids including phosphatidylinositol phosphates, phosphatidylserine, phosphatidic acid and cardiolipin, in the inner leaflet of the cell membrane (*Ding et al., 2016*; *Liu et al., 2016*). In the full-length protein, the C-terminal domain folds back onto the N-terminal domain to auto-inhibit pore formation. Upon linker region cleavage, GSDMs release the active, pore-forming N-terminal domain. The N-terminal domain subsequently oligomerizes on the plasma membrane, assembling into pores with an inner diameter of approximately 10–16 nm, leading to membrane rupture. These pores facilitate the release of pro-inflammatory cytokines, such as IL-1β (4.5 nm) and IL-18 (7.5 nm), thereby amplifying immune signaling. Additionally, pore formation disrupts cellular osmotic balance, resulting in cell swelling, membrane rupture, and pyroptotic cell death. Notably, GSDMD pores exhibit a broader diameter range (10–16 nm), whereas GSDMA3 pores are more uniform (10–14 nm), suggesting potential differences in their functional roles across various cell types (*Ding et al., 2016*; *Liu et al., 2021*).

The lipid-binding active site of GSDMB is located within its N-terminal domain. However, both the full-length GSDMB protein and its N-terminal domain can specifically bind to phosphoinositides and sulfatide, while other members such as GSDMA and GSDMD do not bind to sulfatide (*Chao, Kulakova & Herzberg, 2017*; *Kim et al., 2017*). Unlike other GSDMs, the C-terminal domain of full-length GSDMB does not prevent its binding to phospholipids. This phenomenon may be due to the weaker interdomain interaction between the N- and C-terminal domains of GSDMB compared to other gasdermin family members, or the phospholipid binding site on the N-terminal domain of GSDMB may not be located near the interdomain interface (*Ding et al., 2016*; *Takahashi & Suzuki, 2012*). Sulfatide is synthesized in the Golgi apparatus from galactosylceramide and is subsequently distributed to the cell membrane, lysosomes, and the Golgi apparatus. It is a multifunctional molecule that plays crucial roles in the immune system, nervous system, microbial infections, cancer, insulin secretion, and hemostasis/thrombosis. Elevated levels of sulfatide have been observed on the surface membranes of epithelial cells in various cancers, such as lung adenocarcinoma, renal cancer, and gastric cancer, suggesting that GSDMB may function in the cellular transport of sulfatide. The abundant sulfatides on the surface of cancer cells are natural ligands for P-selectin expressed on endothelial cells and platelets. They can promote cancer cell adhesion and migration, and are associated with the formation of cancer cell emboli, facilitating hematogenous metastasis of cancers (*Chao, Kulakova & Herzberg, 2017*; *Das, Miller & Broide, 2017*; *Garcia, Callewaert & Borsig, 2007*; *Merten et al., 2005*; *Suchanski et al., 2018*; *Suchański & Ugorski, 2016*; *Takahashi & Suzuki, 2012*). In summary, the N-terminal domain of GSDMB can specifically bind to sulfatide, and the overexpression of sulfatide is often associated with cancer progression, suggesting that GSDMB may contribute to cancer cell migration and metastasis.

*Chen et al. (2019)* discovered that full-length GSDMB binds to the caspase-4 recruitment domain, potentially leading to the oligomerization of caspase-4, which in turn induces a conformational change in caspase-4, enhancing its enzymatic activity and promoting the

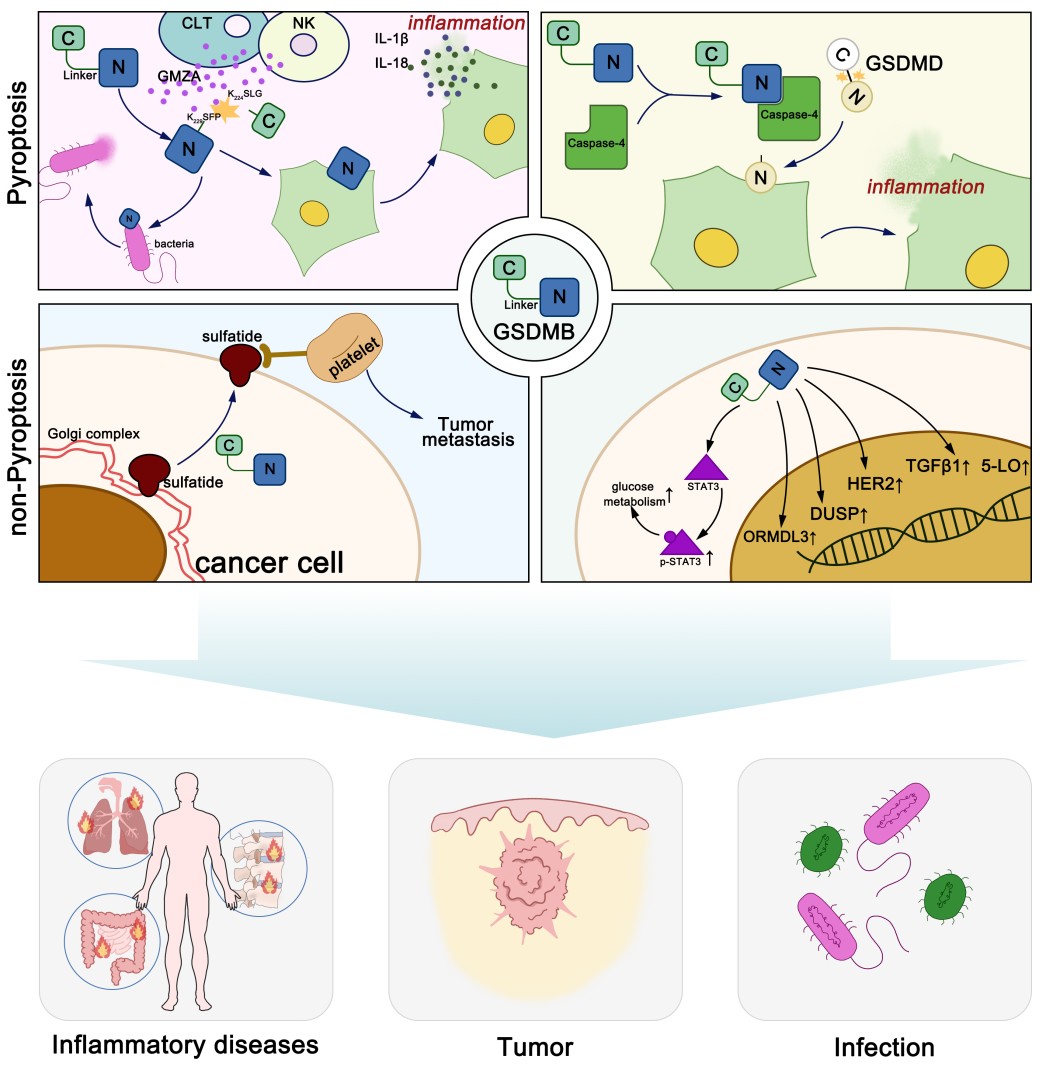

**Figure 1** Molecular functions of GSDMB in pyroptosis and non-pyroptotic cellular processes.

cleavage of GSDMD, thereby inducing non-canonical pyroptosis, a role that can be halted by negative feedback.

In summary, the researchers suggested that the GSDM-N terminus cannot form pores, and the increase in cell death mediated by GSDMB is actually mediated by increased caspase-4 enzymatic activity (*Chen et al., 2019*). However, subsequent reports refuted this hypothesis. In 2020, Zhou et al. reported that natural killer (NK) cells and cytotoxic lymphocytes (CTLs) kill GSDMB-positive cells through pyroptosis. The killing is caused by the cleavage of GSDMB at Lys244/Lys229 by granzyme A (GZMA) secreted by lymphocytes, resulting in a pair of major products (p30 and p16) and another pair of minor cleavage products (p28 and p18). Lys244 is shared by all four isoforms of GSDMB and is the primary physiological cleavage site for GZMA, while Lys229 is present only in GSDMB3-4. This study was the first to demonstrate that GZMA can hydrolyze GSDMs at non-aspartic acid sites

and form pores, redefining the notion that pyroptosis can only be activated by caspases (*Zhou et al., 2020*). *Chen et al. (2019)* constructed various GSDMB fragments based on caspase cleavage sites and demonstrated that none of them exhibited pore-forming activity. Consequently, they shifted their focus to the full-length GSDMB protein, emphasizing its role in modulating caspase-4 activity rather than directly forming membrane pores. In contrast, in the study by *Zhou et al. (2020)* GZMA cleaved GSDMB at different sites, generating an N-terminal fragment with pore-forming activity. Their findings led to the conclusion that GSDMB, once cleaved and activated, also can directly induce pyroptosis.

*Kong et al. (2023)* demonstrated that GSDMB1-4 exhibit distinct functional properties. Specifically, the cleaved N-terminal fragments of GSDMB3 and GSDMB4 can induce pyroptosis, whereas GSDMB1 and GSDMB2 fail to trigger this form of cell death. The non-functional isoforms either lack or possess a modified exon 6, leading to the absence of a stable belt motif, which is thought to facilitate the oligomerization and membrane insertion of GSDMB-NT. Additionally, non-cytotoxic GSDMB-NT can inhibit pyroptosis induced by cytotoxic GSDMB-NT. Upon NK cell attack, cells expressing GSDMB3 undergo pyroptosis followed by further cell death, whereas GSDMB4-expressing cells exhibit a mixed cell death phenotype involving both pyroptosis and apoptosis. In contrast, cells expressing GSDMB1 or GSDMB2 succumb exclusively to apoptosis. Notably, GSDMB4 demonstrates partial resistance to NK cell-mediated cleavage, suggesting that only GSDMB3 retains full functional capacity (*Kong et al., 2023*).

These findings are further supported by *Oltra et al. (2023)* who confirmed that the translation of exon 6 is essential for GSDMB-mediated pyroptosis. Specifically, GSDMB isoforms lacking exon 6 (GSDMB1-2) are incapable of inducing pyroptotic cell death in cancer cells (*Oltra et al., 2023*).

## The relationship between GSDMB and diseases
### GSDMB and cancer

According to statistics from the World Health Organization (WHO), cancer has become the second leading cause of death worldwide, following cardiovascular diseases (*Ahmad & Anderson, 2021*). Due to factors such as global population aging, the incidence of cancer continues to rise, posing a significant challenge to public health. Data from the American Cancer Society (ACS) estimate that in 2024, the United States will report 2,001,140 new cancer cases and 611,720 cancer-related deaths. Although the overall cancer mortality rate has shown a declining trend with the advent of targeted therapies and immunotherapy, the field of cancer treatment still faces multiple challenges, including early diagnosis, drug resistance mechanisms, personalized precision therapy, and the management of treatment-related adverse effects. Therefore, an in-depth investigation into the role and regulatory mechanisms of GSDMB in cancer may provide critical insights for the development of novel therapeutic strategies (*Siegel, Giaquinto & Jemal, 2024*; *Sonkin, Thomas & Teicher, 2024*).

Increasing evidence has confirmed the critical role of GSDMB in cancer; however, few cancer-specific regulatory mechanisms have been identified. Moreover, the conclusions regarding the relationship between GSDMB and cancer are not entirely consistent, as it is

associated with both anticancer and pro-cancer functions, highlighting the heterogeneity of cancers and the complexity of the immune microenvironment. GSDMB exerts its effects in cancers through both pyroptosis-dependent and pyroptosis-independent mechanisms. When GZMA cleaves GSDMB to release the pore-forming N-terminal domain, it can induce cancer cell death through pyroptosis, releasing inflammatory factors and enhancing immune cell infiltration. This may potentially enhance the efficacy of immune checkpoint inhibitors, although it could also lead to the development of a chronic inflammatory microenvironment that may have tumor-promoting effects. However, intact GSDMB can promote various tumorigenic effects, such as invasion, metastasis, and drug resistance (*Hsu et al., 2023*; *Oltra et al., 2023*; *Sarrió et al., 2021*; *Yu et al., 2021*). GSDMB has been implicated in cancer progression, with elevated expression observed in gastric, cervical, breast, and liver cancers (*Carl-McGrath et al., 2008*; *Hergueta-Redondo et al., 2014*; *Sun et al., 2008*). Researchers have demonstrated that GSDMB is located within amplicons, genomic regions frequently amplified during cancer progression. Therefore, GSDMB may contribute to cancer progression and metastasis, although the exact mechanisms remain unclear (*Feng, Fox & Man, 2018*). The dual mechanisms by which pyroptosis-related factors promote and inhibit cancer progression remain to be explored.

The tumor microenvironment (TME) comprises non-malignant cells and their associated components within the tumor, including immune cells, stromal cells, the extracellular matrix (ECM), and various secreted molecules. It plays a crucial role in cancer progression, immune regulation, and therapy resistance (*Liu et al., 2024b*; *Xiao & Yu, 2021*; *Xu et al., 2024*). GSDMB, as an inflammation-related factor, not only regulates the expression of TGF-β1 and 5-lipoxygenase (5-LO) (*Das et al., 2016*), but also undergoes cleavage by GZMA released from NK cells and CTLs to induce inflammatory cell death, thereby impacting the tumor immune microenvironment. Additionally, GSDMB may synergize with immune checkpoint inhibitors to enhance anti-tumor immunity (*Zhou et al., 2020*). In colorectal cancer, GSDMB+ tumor cells have been shown to correlate positively with GZMA+IFN-γ+CD8+ TILs, which enhances tumor immune infiltration (*Yang et al., 2024*). In melanoma, the tumor may reduce the expression of pyroptosis-related genes (PRGs) such as *GSDMB* in immune cells, thereby lowering pyroptotic activity and inhibiting anti-tumor immunity (*Zhang et al., 2023*). Furthermore, a PRG-based prognostic models, including *GSDMB*, serve as independent prognostic factors for adrenal cortical carcinoma (ACC) and are closely associated with immune cell infiltration, tumor mutation burden, microsatellite instability, and immune checkpoints (*Gao et al., 2023*). In summary, due to the complexity and individual specificity of the TME, GSDMB, through its interactions, may play differential roles in various cancers by affecting immune evasion, immune therapy responses, and other aspects.

*Kong et al. (2023)* found that GSDMB1-3 are the most abundant isoforms in the tested tumor cell lines. In bladder cancer and cervical cancer, the expression of cytotoxic GSDMB3-4 is associated with better prognosis, while GSDMB1-2, frequently upregulated in tumors, are not associated with better prognosis, indicating that GSDMB3-4 mediated pyroptosis has a protective role in these tumors (*Kong et al., 2023*). *Oltra et al. (2023)* reported that GSDMB2 expression, rather than exon 6-containing isoforms (GSDMB3-4), is associated

with poor clinical-pathological parameters in breast cancer. Specific residues in exon 6 and other regions of the N-terminal domain are crucial for GSDMB-triggered cell death and mitochondrial damage. Additionally, specific proteases (granzyme A, neutrophil elastase, and caspases) differentially regulate pyroptosis by cleaving GSDMB. Granzyme A from immune cells can cleave all GSDMB isoforms, but only in exon 6-containing isoforms does this process lead to the induction of pyroptosis. Conversely, GSDMB isoforms cleaved by neutrophil elastase or caspases produce non-cytotoxic short N-terminal fragments, indicating that these proteases may act as inhibitors of pyroptosis (*Oltra et al., 2023*). The expression of different GSDMB isoforms in various cancers and how GSDMB influences cancer biology remain unclear. *Das et al. (2016)* discovered that GSDMB regulates TGF-β1 expression through nuclear localization and transcriptional regulatory effects, suggesting that its function may largely be independent of its pore-forming activity. This finding further raises the possibility that other GSDMs may also possess unknown functions that are independent of their cleavage or pore-forming abilities (*Das et al., 2016*; *Liu et al., 2021*).

### Breast cancer

Breast cancer is one of the most common malignancies. In breast cancer, high GSDMB expression has been associated with cancer progression, and patients with high GSDMB expression have shown poor responses to HER2-targeted therapy. In HER2-positive breast cancer, increased *GSDMB* gene expression is associated with poor prognosis (*Hergueta-Redondo et al., 2014*), characterized by shorter survival and higher metastasis rates, as well as adverse responses to HER2-targeted therapy. Recent studies have shown that GSDMB reduces the sensitivity to HER2-targeted therapy by promoting protective autophagy and enhancing autophagosome-lysosome fusion through its interaction with Rab7/LC3B. Additionally, research indicates a significant co-expression trend between *GSDMB* and *ERBB2*. The *GSDMB* gene is located 175 kilobases downstream of *ERBB2*, and its gene amplification is highly correlated with *HER2 (ERBB2)* gene amplification. GSDMB overexpression occurs in approximately 60% of HER2-positive breast cancer cases. *Molina-Crespo et al. (2019)* demonstrated that targeting GSDMB with antibodies effectively reduces metastasis, invasion, and therapy resistance in HER2-positive breast cancer (*Edgren et al., 2011*; *Gámez-Chiachio et al., 2022*; *Hergueta-Redondo et al., 2014*; *Hergueta-Redondo et al., 2016*). Several splice isoforms of GSDMB exist in humans, and GSDMB2 is strongly associated with the tumorigenic and metastatic phenotype of breast cancer cells (*Hergueta-Redondo et al., 2016*). This suggests that GSDMB may serve as a novel prognostic marker for breast cancer;

### Gastric cancer

Gastric cancer is a malignancy originating from gastric cells, characterized by poor prognosis and high mortality. One study indicated that *GSDMA* acts as a tumor suppressor gene in gastric cancer, while *GSDMB* is overexpressed in some gastric cancer cells and may function as an oncogene. *Komiyama et al. (2010)* examined *GSDMB* expression in normal and cancerous gastric tissues and found that *GSDMB* is highly expressed in most gastric cancer tissues but not in most normal gastric tissues, possibly related to gastric cancer invasion.

They also identified an Alu element, approximately 300 bp in length, belonging to the short interspersed nuclear elements (SINE) family of retrotransposons, located upstream of *GSDMB*. The study found that the Alu element regulates *GSDMB* expression in cancer cells, possibly involving the IKZF (zinc finger transcription factor) binding motif present in this element, which plays a key role in upregulating *GSDMB* expression. Another study by *Saeki et al. (2015)* found that *GSDMB* expression is driven by two promoters, including a cellular promoter and an LTR-derived promoter. These studies suggest that *GSDMB* expression levels and the activation of Alu elements and LTR-derived promoters may serve as useful molecular markers for assessing gastric cancer development and progression (*Komiyama et al., 2010*; *Saeki et al., 2015*). *Carl-McGrath et al. (2008)* found that *GSDMB* may also be overexpressed in liver cancer and colorectal cancer tissues (*Saeki et al., 2009*).

### Cervical cancer

The rs8067378 SNP variant is located downstream of *GSDMB*, corresponding to the cellular promoter and LTR region, and increases *GSDMB* expression, which is associated with the progression of cervical squamous cell carcinoma. Therefore, *GSDMB* expression may be a risk factor for cervical squamous cell carcinoma and may promote cancer cell growth and accelerate metastasis of low-grade cancer cells (*Lutkowska et al., 2017*; *Sun et al., 2008*).

### Colorectal cancer

*Sun et al. (2024)* found that the expression pattern of GSDMB is related to the prognosis, progression, and immune response of colorectal cancer (CRC). The study revealed that GSDMB is significantly expressed in CRC tissues, and cytoplasmic GSDMB expression serves as an independent favorable prognostic indicator. Moreover, CRC cells with high GSDMB expression show increased sensitivity to 5-fluorouracil chemotherapy. Additionally, GSDMB expression is associated with systemic inflammation markers such as neutrophils and lymphocytes in peripheral blood (*Sun et al., 2024*). *Jiang et al. (2024)* found that GSDMB regulates the expression of DUSP6 through interaction with IGF2BP1, thereby influencing the ERK signaling pathway, inhibiting cell proliferation, and promoting cell death. *Yang et al. (2024)* explored the role of CD8+ tumor-infiltrating lymphocytes (TILs) co-expressing GZMA and interferon-γ (IFN-γ) in regulating GSDMB expression in the TME of colorectal cancer. Their research showed a positive correlation between GSDMB+ tumor cells and GZMA+IFN-γ+CD8+TILs, and high infiltration of GZMA+IFN-γ+CD8+TILs is associated with better patient prognosis (*Yang et al., 2024*). These studies suggest that GSDMB not only has a potential inhibitory role in the progression of CRC but also influences the immune microenvironment, making it a potential biomarker for disease prognosis and a target for therapeutic intervention;

### Lung cancer

*Liang et al. (2024)* explored the complex role of GSDMB in lung cancer and other respiratory diseases. New evidence suggests that promoting GSDMB-mediated pyroptosis can inhibit tumor growth and reverse resistance. In lung cancer, particularly non-small cell lung cancer (NSCLC), GSDMB-induced pyroptosis has been shown to inhibit tumor cell proliferation and metastasis. However, pyroptosis can also lead to adverse effects associated

with tumor therapy, such as increased inflammation and tissue damage. This indicates the dual role of GSDMB in promoting and inhibiting tumor growth, requiring a balance between its anti-tumor benefits and potential side effects in tumor therapy (*Liang et al., 2024*).

### Bladder cancer

In bladder cancer, the role of GSDMB remains controversial. *He et al. (2021)* found that GSDMB expression is higher in bladder cancer tissues compared to adjacent normal tissues. Their study demonstrated that GSDMB promotes bladder cancer progression through interaction with STAT3, enhancing STAT3 phosphorylation and modulating glucose metabolism. This pathway activates tumor cell growth and invasion, suggesting that *GSDMB* may function as an oncogene in bladder cancer. Furthermore, they identified that USP24 stabilizes GSDMB, preventing its degradation and thereby further promoting tumor growth. In contrast, *Wang et al. (2023)* focused on the therapeutic potential of Anlotinib, a multi-target tyrosine kinase inhibitor, in treating GSDMB-positive bladder cancer. Their bioinformatics analysis revealed that patients with high GSDMB expression had better overall survival, and that GSDMB expression was significantly elevated in tumor tissues compared to normal tissues. Further investigations demonstrated that Anlotinib treatment enhanced the secretion of antitumor factors and effectively reduced tumor growth in GSDMB-positive bladder cancer.This discrepancy may be attributed to the dual role of GSDMB, where under certain conditions, it promotes tumor growth (*e.g.*, *via* STAT3 signaling), whereas in the context of treatment, it may enhance therapeutic efficacy and regulate immune responses. However, the lack of further validation of GSDMB subtypes in these studies could be one of the reasons for the conflicting results observed. Therefore, further research is required to reconcile these mechanisms and comprehensively understand the role of GSDMB in bladder cancer progression and therapeutic response (*He et al., 2021*; *Wang et al., 2023*).

### Kidney cancer

*Cui et al. (2021)* retrieved the transcriptional expression profiles of *GSDMB* in clear cell renal cell carcinoma (ccRCC) tissues and normal tissues from The Cancer Genome Atlas (TCGA) database and validated them in the Gene Expression Omnibus (GEO) database. Using various bioinformatics analyses, they determined the relationship between *GSDMB* mRNA expression and immune infiltration. They demonstrated that GSDMB mRNA and protein expression are upregulated in ccRCC and positively correlated with higher TNM stages, suggesting that GSDMB may be a potential biomarker associated with poor prognosis and may have unique functions in regulating immune infiltration in ccRCC. *Huang et al. (2024)* found that gasdermin (GSDM) family genes play an important role in ccRCC. Using multiple bioinformatics databases, such as TCGA, GEPIA, Metascape, and cBioPortal, they analyzed gene differential expression, gene mutations, and their impact on prognosis and immune regulation. The results showed that *GSDMA*, *GSDMB*, *GSDMC*, and *GSDMD* mRNA expression are upregulated in ccRCC tissues. High expression levels of *GSDMB*, *GSDMD*, and *DFNA5* are associated with poorer pathological features and lower survival rates in ccRCC patients. In particular, GSDMB was identified as an independent prognostic

marker, indicating its potential as a therapeutic target and biomarker for ccRCC (*Cui et al., 2021*; *Huang et al., 2024*). While TCGA and other public databases provide valuable resources for transcriptomic analysis, recent studies have emphasized potential limitations of bulk RNA-seq data. These include technical biases (such as batch effects and sequencing depth variation) and biological confounders (such as tumor heterogeneity and immune cell infiltration), which may affect the accuracy of gene expression estimates (*Liu, Guo & Wang, 2024a*; *Liu et al., 2025*). Therefore, findings based on TCGA data, including the expression and prognostic value of GSDMB, should be interpreted with caution, and ideally validated using complementary methods such as single-cell or spatial transcriptomics.

## GSDMB and autoimmune and inflammatory diseases

In recent years, multiple studies have also revealed that GSDMB plays a crucial role in the onset and development of various inflammatory diseases and autoimmune disorders. SNPs of the *GSDMB* gene have shown significant associations with multiple diseases by affecting pyroptosis and inflammatory responses:

### Diabetes mellitus

*GSDMB* gene is located within the 17q21 region, which harbors multiple genes associated with autoimmune diseases, including *ORMDL3*, a gene that has been demonstrated to regulate *GSDMB* expression. Several single nucleotide polymorphisms (SNPs) within this region, such as rs12150079, rs2305480, and rs3894194, have been significantly associated with type 1 diabetes (T1D) (*Barrett et al., 2009*; *Witsøet al., 2015*). The risk allele of SNP rs12936231 has been linked not only to an increased susceptibility to asthma and T1D but also to upregulated expression of *GSDMB* and *ORMDL3*, along with downregulation of ZPBP2. Inflammation is a key contributor to T1D pathogenesis, and GWAS findings suggest that SNPs within the 17q21 region may influence *GSDMB* expression, thereby modulating immune regulation and disease susceptibility in T1D (*Barrett et al., 2009*; *Verlaan et al., 2009a*). Additionally, the N-terminal domain of GSDMB can interact with cell membrane components to form pores, leading to the release of cellular contents and pyroptosis, which may play a key role in the immunopathology of T1D (*Feng, Fox & Man, 2018*).

### Ankylosing spondylitis

In ankylosing spondylitis, variations in the *GSDMB* gene also show significant impact. *Qiu et al. (2013)* found that certain SNPs in *GSDMB* are associated with susceptibility and severity of ankylosing spondylitis in the Chinese Han population. These SNPs may affect the immune system's response to inflammatory signals by altering *GSDMB* expression and function (*Qiu et al., 2013*). Specifically, the N-terminal domain of GSDMB can trigger pyroptosis in immune cells, releasing pro-inflammatory cytokines such as IL-1β and IL-18, thereby exacerbating chronic inflammation in ankylosing spondylitis (*Feng, Fox & Man, 2018*).

### Asthma

GSDMB is highly expressed in the bronchial epithelium of asthma patients and has been shown to induce the expression of TGF-β1 (*Das et al., 2016*). *Bouzigon et al. (2008)* showed

that GSDMB is significantly associated with asthma and asthma-related phenotypes, such as bronchial hyperresponsiveness (BHR) and total IgE levels. The *GSDMB* gene influences asthma occurrence and development by encoding proteins involved in cell differentiation, cell cycle regulation, and apoptosis. Studies have found that single nucleotide polymorphisms (SNPs) in *GSDMB*, particularly rs7216389, are closely related to asthma susceptibility. This SNP is located in the intronic region of the *GSDMB* gene and influences the transcriptional level of the *ORMDL3* gene, further regulating airway smooth muscle remodeling and fibrosis, increasing bronchial hyperresponsiveness. *Li et al. (2021)* conducted a GWAS study and identified several SNPs of GSDMB (*e.g.*, rs1031458 and rs3902920) that modulate the expression of GSDMB and are significantly associated with asthma severity and long-term exacerbations. Furthermore, GSDMB expression levels were found to be positively correlated with the expression of genes involved in multiple antiviral pathways, suggesting that viral infections and the activation of antiviral pathways may contribute to the development of severe asthma and asthma exacerbations (*Li et al., 2021*). Additionally, SNP rs11078928 has been found to regulate the transcription of GSDMB pyroptosis-related isoforms (GSDMB3-4) (*Morrison et al., 2013*), suggesting that excessive pyroptosis may play a pathogenic role in asthma. In studies across multiple ethnic groups, such as Puerto Ricans, African Americans, and Mexicans, rs7216389 has shown a significant association with asthma (*Galanter et al., 2008*; *Zhao et al., 2015*). A study of Korean children also found that asthmatic children carrying the *GSDMB* rs7216389 TT genotype had significantly elevated total IgE levels and bronchial hyperresponsiveness, further confirming the role of the *GSDMB* gene in asthma (*Yu et al., 2011*). Additionally, the association between *GSDMB* and the *ORMDL3* gene is particularly significant in early-onset asthma patients exposed to tobacco smoke, suggesting that these two genes may be co-regulated and highlighting the regulatory role of environmental factors in the impact of the GSDMB gene on asthma occurrence (*Bouzigon et al., 2008*).

### Inflammatory bowel disease

The SNPs in the *GSDMB* gene are associated with susceptibility to inflammatory bowel disease (IBD). In contrast to asthma, studies show that SNP risk alleles for IBD typically downregulate *GSDMB* expression in gut/immune cells. The rs2872507 risk allele is significantly associated with decreased *GSDMB* expression levels in intestinal tissues, a reduction observed in both inflamed and non-inflamed mucosal tissues. Additionally, this risk allele is associated with increased expression of *GSDMA* and *LRRC3C* genes and decreased expression of *PGAP3* and *ZPBP2* genes. These gene expression changes may impact apoptosis and proliferation, further modulating the pathophysiological processes of IBD (*Söderman, Berglind & Almer, 2015*; *Verlaan et al., 2009b*). Nitish Rana and colleagues found that GSDMB is primarily expressed in intestinal epithelial cells (IEC), and within these cells, GSDMB does not trigger the typical pyroptotic response. The presence of IBD-associated *GSDMB* SNPs leads to functional defects, impairing epithelial restitution/repair. They also observed that the immunosuppressant methotrexate can upregulate the expression of uncleaved GSDMB in intestinal epithelium, suggesting GSDMB as a potential therapeutic target for IBD, particularly in regulating epithelial barrier

function and attenuating inflammation (*Rana et al., 2022*). *Chen et al. (2019)* proposed that in IBD, GSDMB does not directly trigger pyroptosis by cleaving and releasing the N-terminal domain but rather promotes non-canonical pyroptosis through direct interaction with caspase-4. Although many studies have demonstrated the association of GSDMB with IBD susceptibility, exploration of the mechanisms by which GSDMB influences IBD remains relatively scarce. Whether pyroptosis induced by GSDMB affects the onset and progression of intestinal inflammation is still to be clarified;

### Psoriasis

Psoriasis is a common chronic skin disease characterized by excessive proliferation and inflammation of skin cells. *Nowowiejska et al. (2024)* showed that GSDMB expression levels in serum, urine, and skin tissues of psoriasis patients were significantly higher than in control groups without skin diseases. GSDMB expression in psoriatic plaques was mainly concentrated in the dermis and epidermis, and its expression was significantly increased in psoriatic plaques compared to non-lesional skin and healthy controls. These findings suggest that GSDMB may play an important role in the onset and development of psoriasis by regulating keratinocyte proliferation and migration (*Nowowiejska et al., 2024*).

### Chronic rhinosinusitis

The SNPs of the *GSDMB* gene are significantly associated with chronic rhinosinusitis (CRS). Studies have shown that individuals carrying the *GSDMB* rs7216389 SNP exhibit a higher susceptibility to CRS. In a multi-center retrospective case-control study, participants from two otolaryngology centers at the University of Arizona and the University of Pennsylvania were included. The results revealed that individuals carrying the *GSDMB* rs7216389 risk allele were more likely to develop CRS in both populations. Furthermore, the study found that *GSDMB* rs7216389 may promote the development of CRS by affecting the inflammatory response and remodeling processes in the nasal cavity and airways. The research also noted that asthma and CRS share common pathophysiological mechanisms and may manifest as unified airway disease. Rhinovirus (RV) infection plays a key role in this process, as it exacerbates the symptoms of both asthma and CRS. The *GSDMB* rs7216389 SNP is also associated with abnormal immune responses to RV infection, which may contribute to the onset and exacerbation of asthma and CRS. These findings suggest a potential role of the *GSDMB* gene in CRS and related upper airway diseases, which may provide new directions for future treatments of these conditions (*Zack et al., 2021*).

### Infectious diseases

GSDMB plays a key role in immune responses to a variety of infectious diseases. Upon activation, GSDMB not only targets bacterial membranes to induce bacterial lysis, but also activates specific pathways involved in virus-induced cell death and inflammation. Additionally, *GSDMB* gene is closely linked to the activation of various immune-related genes, which can aid in predicting the severity and prognosis of infectious diseases (*Hansen et al., 2021a*; *Li et al., 2023*; *Miranzadeh Mahabadi et al., 2024*; *Pasanen et al., 2024*).

*Hansen et al. (2021a)* explored the role of GSDMB in NK cell-mediated antibacterial defense. Their study indicated that once activated, GSDMB directly dissolves bacteria by

recognizing and binding to specific bacterial lipids, without the need for host cell death. Further research showed that the N-terminal domain of GSDMB binds to phospholipids on the membranes of Gram-negative bacteria, forming pores that induce bacterial death. This finding suggests that GSDMB has a specialized antimicrobial function, particularly in membrane binding and bacterial dissolution. *Shigella flexneri* (enteroinvasive Shigella) targets GSDMB by secreting the effector protein IpaH7.8, which ubiquitinates and degrades GSDMB, thereby suppressing its function to evade NK cell-mediated bacterial clearance, revealing a bacterial strategy to evade host immune responses (*Hansen et al., 2021a*).

*Li et al. (2023)* found that GSDMB plays a dual role in sepsis: it facilitates bacterial clearance through non-canonical pyroptosis, while also regulating this process through caspase-7 to prevent excessive inflammation. GSDMB promotes non-canonical pyroptosis by interacting with caspase-4, but during apoptosis, activated caspase-7 cleaves GSDMB at the D91 site, blocking its role in non-canonical pyroptosis, thus inhibiting excessive inflammation and serving a protective role in sepsis. In sepsis mouse models, caspase-7 inhibition or deficiency in GSDMB transgenic mice resulted in more severe disease phenotypes, further proving the significance of the caspase-7/GSDMB axis in sepsis. This finding provides new potential therapeutic targets for sepsis, particularly in balancing pyroptosis and apoptosis (*Li et al., 2023*).

Acute viral bronchiolitis is a common lower respiratory infection (LRI) and a major cause of infant hospitalization worldwide (*Florin, Plint & Zorc, 2017*). *Pasanen et al. (2024)* conducted a GWAS study to investigate the genetic factors contributing to bronchiolitis susceptibility, revealing several key genetic loci. Among them, variations within the *GSDMB* gene locus were found to be significantly associated with susceptibility to viral bronchiolitis, particularly cases caused by non-respiratory syncytial virus (non-RSV). These findings suggest that GSDMB plays a crucial role in immune responses, especially against viral respiratory infections. Furthermore, studies have shown that severe bronchiolitis in early childhood, particularly when caused by non-RSV viruses, is associated with an increased risk of developing asthma later in life (*Meissner, 2016*). The *GSDMB* gene is known to be associated with asthma susceptibility. These findings further elucidate the genetic link between early childhood respiratory infections and the development of asthma, indicating that GSDMB may serve as a key biomarker for bronchiolitis severity and asthma susceptibility (*Pasanen et al., 2024*).

*Miranzadeh Mahabadi et al. (2024)* investigated the interaction between monkeypox virus (MPXV) and GSDMB in neural cells, finding that MPXV infection triggers pyroptosis through GSDMB. MPXV preferentially infects human astrocytes, inducing immune responses and activating inflammation-related genes. In this process, MPXV specifically induces proteolytic cleavage of GSDMB, leading to plasma membrane rupture and cell death (pyroptosis), which may contribute to neurological symptoms observed in monkeypox patients. Moreover, the study showed that dimethyl fumarate (DMF) could inhibit GSDMB cleavage induced by MPXV infection, reducing cytotoxic responses, thereby providing a new therapeutic approach for treating monkeypox-related neurological diseases (*Miranzadeh Mahabadi et al., 2024*).

GSDMB not only plays a critical role in the immune response and pathogenesis of infectious diseases, but also provides a new biomarker for predicting the prognosis of these diseases. These findings open up new directions for developing therapeutic strategies targeting GSDMB.

## ADVANCES IN TUMOR THERAPY INVOLVING GSDMB

In tumors, the pro-tumor or anti-tumor effects of GSDMB may depend on the biological context of the tumor (*Sarrió et al., 2021*). In breast cancer and gastric cancer, *GSDMB* is often co-expressed with the *HER2/ERBB2* oncogene, and *GSDMB* overexpression promotes tumor invasion, metastasis, and treatment resistance (*Gámez-Chiachio et al., 2022*; *Hergueta-Redondo et al., 2014*; *Molina-Crespo et al., 2019*; *Sarrio et al., 2022*). Targeting GSDMB with antibodies has been shown to reduce its pro-tumor functions (*e.g.*, cell migration, metastasis, and drug resistance) in breast cancer cells (*Molina-Crespo et al., 2019*). At the same time, if the pore-forming pyroptotic function of GSDMB is activated in cancer cells, it may also have anti-tumor effects, which can be triggered *in vitro* by granzyme A (GZMA) cleavage. In the context of immune activation, CTLs and NK cells first recognize and engage with target cells, such as cancer cells. They then release perforin and granzymes. Perforin forms pores in the target cell membrane, allowing granzyme A to enter the target cell. Once inside the target cell, granzyme A specifically cleaves GSDMB at the Lys229/Lys244 sites, releasing the GSDMB-N terminal domain with pore-forming activity. This domain oligomerizes on the target cell membrane to form pores, inducing pyroptosis and subsequent antitumor immune responses (*Zhou et al., 2020*). Therefore, triggering GSDMB pyroptosis is considered a promising method for effectively killing tumors. However, to develop future therapies targeting GSDMB, the precise functional domains and regulatory mechanisms of GSDMB pyroptosis must be thoroughly defined, as there are currently many controversial and contradictory results (*Oltra et al., 2023*). *Zhou et al. (2020)* found that while GSDMB expression does not affect the growth of colorectal cancer or melanoma in mice, GSDMB expression synergistically enhances tumor growth inhibition when combined with anti-PD1 checkpoint inhibitors. In another study, the researchers established a bioorthogonal chemical system in which the tumor imaging probe phenylalanine trifluoroborate (Phe-BF3) can enter cells and desilylate, cleaving a designed linker containing a silyl ether group. This system allows controlled release of drugs from antibody-drug conjugates in mice. When combined with nanoparticle-mediated delivery, Phe-BF3-mediated desilylation can selectively release client proteins (including active gasdermin) into tumor cells in mice. The application of this bioorthogonal system to breast tumors showed that inducing pyroptosis in tumor cells significantly inhibits breast tumor growth. The system's application suggests that inflammation triggered by pyroptosis can elicit a powerful anti-tumor immune response and synergize with immune checkpoint inhibitors, potentially improving immunotherapy responsiveness (*Wang et al., 2020*).

In gastric cancer, particularly in HER2-positive gastric cancer, GSDMB plays a crucial role in pyroptosis. Lin et al. demonstrated that the bispecific antibody IBI315, which

simultaneously targets PD-1 and HER2, significantly enhances tumor cell killing by inducing GSDMB-mediated pyroptosis. Specifically, IBI315 activates the cleavage of GSDMB, leading to the release of inflammatory factors such as IL-18 from tumor cells. These factors further activate T cells, and the activation of T cells, in turn, enhances GSDMB expression through a positive feedback mechanism, forming a tumor cell killing loop. The upregulation of GSDMB in HER2-positive gastric cancer cells provides a theoretical basis for the efficacy of IBI315, suggesting the potential application of GSDMB in gastric cancer immunotherapy (*Lin et al., 2023*).

In lung adenocarcinoma (LUAD), the combination of inetetamab (an anti-HER2 monoclonal antibody) and cisplatin significantly enhances the antitumor effect, primarily through the induction of NLRP3/caspase-1/GSDMB-mediated pyroptosis. Cui et al. found that when inetetamab was used in combination with cisplatin, it inhibited the HER2/AKT/Nrf2 signaling pathway, increased ROS levels, and activated the NLRP3 inflammasome, which in turn triggered the cleavage of GSDMB. This process led to the rupture of tumor cell membranes, releasing pro-inflammatory factors such as HMGB1, and promoting the activation of immune cells. The study indicates that this pyroptosis induction not only improves the efficacy of cisplatin but also enhances the sensitivity of cisplatin-resistant lung adenocarcinoma cells (*Cui et al., 2023*).

*Kong et al. (2023)* demonstrated that GSDMB splice variants differ in function. The N-terminal fragments of GSDMB3 and GSDMB4 can induce pyroptosis, while GSDMB1 and GSDMB2 do not. Tumors may block and evade killer cell-triggered pyroptosis by producing non-cytotoxic GSDMB isoforms. In the limited current research, the expression of the cytotoxic GSDMB3-4 isoforms in bladder cancer and cervical cancer is generally associated with a better prognosis (*Kong et al., 2023*), whereas the GSDMB2 isoform is strongly correlated with the tumorigenic and metastatic phenotypes of breast cancer cells (*Hergueta-Redondo et al., 2016*). Therefore, therapies that selectively produce cytotoxic GSDMB isoforms through alternative splicing may improve anti-tumor immunity.

## CONCLUSIONS

In summary, GSDMB plays a pivotal role in the initiation and progression of both tumor and non-tumor diseases. Beyond its diverse functions in cancer, GSDMB has garnered significant attention for its involvement in inflammatory diseases, including asthma, inflammatory bowel disease, and viral infections. Through mechanisms such as immune modulation, pyroptosis regulation, and epithelial repair, GSDMB can have distinct and even opposing effects depending on the disease context, contributing to both disease progression and resolution based on the cellular environment.

Given its complex and context-dependent nature, GSDMB exemplifies a "double-edged sword." On one hand, it can exert tumor-suppressive effects, as demonstrated in gastric and lung cancers, where GSDMB synergizes with immune checkpoint inhibitors and cisplatin to enhance anti-tumor responses. Additionally, GSDMB is implicated in epithelial repair and inflammation regulation, particularly in conditions like IBD, offering potential for therapeutic benefit. On the other hand, certain SNPs in the GSDMB gene have been

associated with cancer progression and drug resistance, as observed in breast cancer, as well as the exacerbation of inflammatory diseases, including asthma and chronic rhinosinusitis . Understanding how different GSDMB isoforms function in various diseases is crucial for obtaining a comprehensive view of its functional diversity. Moreover, GSDMB's role in modulating immune responses and its interactions within the tumor microenvironment position it as a promising therapeutic target. Targeting GSDMB could enhance its tumor-suppressive effects in cancer while alleviating chronic inflammation in diseases like asthma and IBD.

Recent advances in liquid biopsy technologies, including circulating tumor DNA (ctDNA) analysis and epigenetic profiling, offer promising avenues for the noninvasive detection of disease-associated molecular features (*Gonzalez et al., 2024*; *Jahangiri, 2024*; *Ohyama et al., 2024*). Although not yet widely applied to the gasdermin family, these tools may provide future opportunities to monitor GSDMB-associated molecular alterations and support its potential as a diagnostic or prognostic biomarker.

Future research should focus on developing specific GSDMB-targeted therapies, optimizing their efficacy, and minimizing associated risks. Investigating the molecular mechanisms underlying GSDMB's diverse functions will be essential for refining therapeutic strategies and improving patient outcomes across a wide range of diseases.

### Funding
This work was supported by the Sichuan Science and Technology Program [2022YFS0133]. The funders had no role in study design, data collection and analysis, decision to publish, or preparation of the manuscript.

### Grant Disclosures
The following grant information was disclosed by the authors:
The Sichuan Science and Technology Program: 2022YFS0133.

### Competing Interests
The authors declare there are no competing interests.

### Author Contributions

- Weixiao Yang conceived and designed the experiments, performed the experiments, analyzed the data, prepared figures and/or tables, authored or reviewed drafts of the article, and approved the final draft.
- Xu Hu conceived and designed the experiments, performed the experiments, analyzed the data, authored or reviewed drafts of the article, and approved the final draft.
- Xiang Li conceived and designed the experiments, authored or reviewed drafts of the article, and approved the final draft.

### Data Availability
This is a literature review.

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
