# Peer review of "Unveiling the role of gasdermin B in cancer and inflammatory disease: from molecular mechanisms to therapeutic strategies"

_PeerJ, doi:10.7717/peerj.19392_

## Round 0.1 · original submission · Major Revisions

The reviewers acknowledge the relevance and significance of the manuscript, as GSDMB plays a crucial role in cancer and inflammatory diseases. However, they find that the current review lacks novelty and differentiation from previous literature. To strengthen the manuscript, the authors should provide a clearer justification for studying GSDMB beyond its uniqueness in being absent in rodents. Emphasizing its genetic association with diseases, particularly through GWAS studies, would add depth and context.
A key concern raised is the need for a more thorough and accurate literature review. The authors should expand their search methodology to include broader MeSH terms (e.g., Gasdermin B, GSDML, allergic disease) and ensure proper citation of sources. Several references are misattributed (e.g., phosphoinositide binding study cited incorrectly).
The dual role of GSDMB in cancer progression—both pro-tumor and anti-tumor—requires greater clarification. Reviewers recommend elaborating on the tumor microenvironment, including GSDMB’s interaction with immune and stromal cells, and discussing recently discovered cell death pathways (ferroptosis, cuproptosis, disulfidptosis) in the context of GSDMB. The functional distinctions between GSDMB isoforms should also be explored in greater detail, particularly regarding their role in cancer progression and therapeutic responses.
The section on inflammatory diseases is underdeveloped. Reviewers suggest expanding the discussion on asthma, inflammatory bowel disease, and chronic sinusitis, incorporating both GWAS data and mechanistic insights.
The conclusion should offer a more balanced discussion on the controversial aspects of GSDMB’s function in disease, addressing conflicting data in the literature. The "double-edged sword" nature of GSDMB should be supported with specific examples.
Reviewers also recommend adding a figure to illustrate GSDMB’s cellular functions and improving the consistency of terminology (e.g., standardizing how GSDMB isoforms are labeled). Additionally, the manuscript would benefit from thorough proofreading to correct minor grammatical and typographical errors.
Overall, the reviewers emphasize that while the topic is important, substantial revisions are required to improve clarity, depth, and citation accuracy. Strengthening these aspects will make the manuscript more impactful and distinguish it from previous reviews.

Reviewer 1 ·

Basic reporting

Yang et al submit a manuscript for potential publication in PeerJ about "Unveiling the Role of Gasdermin B in Cancer and Inflammatory disease: From Molecular Mechanisms to Therapeutic Strategies." Atlhough there are already many literature reviews on GSDMB and diseases including cancer and inflammatory diseases, this remains an interesting topic considering the strong genetic association of GSDMB with diseases such as asthma and cancer but whose mechanisms remain undefined. However, the current review needs to be improved with regards to citations, survey methodology as well as writing. The comments below are meant to improve the current manuscript.

Considering the strong association of GSDMB with many diseases as well as its role in cell proliferation and cell death, the topic of this review is broad and cross-disciplinary and within the scope of PeerJ.

There have already been many reviews on GSDMB and/or pyroptosis in cancer and inflammatory disease (PMID: 32431546; PMID: 35821185; PMID: 37997534; PMID: 35580164 to name a few) and the current review does not quite differentiate itself from the others. Still, elucidating the functional role of GSDMB in disease is an exciting research area and it is suggested that the authors present new/different perspectives on this topic.

The Introduction needs to be improved. It does not give much emphasis on why research on GSDMB is important, other than, as the authors wrote, “GSDMB is unique as it is the only GSDM gene not found in rodents…” and “although GSDMB has been associated with various human diseases, its precise mechanisms and functions remain to be fully understood.” It sounds like GSDMB is just another gasdermin that is less explored, and should be studied. Epidemiological studies (e.g. GWAS) had revealed the association of GSDMB with human diseases especially asthma (since 2007 PMID: 20860503; PMID: 17611496 and many other GWAS studies on GSDMB and asthma and other allergic and inflammatory disease) long before the gasdermins were identified as pyroptosis executioners in 2015 - this adds to the significance of studying the basic molecular/cellular function as well as the mechanistic association of GSDMB with diseases.

Experimental design

For the survey methodology used in this manuscript, this reviewer believes that additional keywords such as Gasdermin B, GSDML, and allergic disease should be used in the MeSH terms. There are some GSDMB-related papers that are important but were not included because these terms were not included.

Some of the sources are not adequately cited. For example, lines 125-126 which refer to binding of GSDMB to phosphoinositides and sulfatides, were not properly cited. This research was from Chao et al 2017 and not Kim et al 2017. Please cite the appropriate paper. There are other missing citations in the additional comments.

Validity of the findings

The conclusion mostly focuses on the role of GSDMB in cancer or tumorigenic and leaves out its role in other diseases or conditions (for example, inflammatory states) although a significant portion of the body of the review talks about "GSDMB and genetic and (auto)inflammatory diseases."

Additional comments

In Line 40, the authors wrote GSMDB1-4 as corresponding homologs in mice – this is not correct as GSDMB gene is not found in mouse or rat genome.

In line 94, the authors write that GSDMB protein consists of 411 amino acids. The authors should update this information. The longest isoform is 416 amino acids (GSDMB3) while GSDMB1 which lacks exon 6 (13 amino acids) is 403 amino acids. The presence of exon 6 is important in the pyroptosis-inducing activity of GSDMB; this exon 6 is deleted in GSDMB1 and it was suggested that a splice variant (or SNP) abolishes the ability of GSDMB to act as pyroptosis executioner via its N-terminal fragment. Hence, it is important to report our current updated knowledge on GSDMB.

In line 22, the authors write that gasdermin pores are 1-2 um pores. Is this diameter, circumference or radius? Also, please verify the citation as the size of gasdermin pores are way smaller than this. Also, did Liu et al 2021 mention the size of pores to be this big? Proper citation should be used.

In lines 125-126 which refer to binding to phosphoinositides and sulfatides, this is not properly cited. This research was from Chao et al 2017 and not Kim et al 2017. Please cite the appropriate paper.

Lines 128-134 do not have the proper citations.

Can the authors comment on why there was difference in the results of Chen et al 2019 compared to the other papers that showed pyropotisis by GSDMB-N terminal fragment (especially the earlier papers on GSDMB and pyroptosis)?

In Line 174, it is suggested to just use the Kong et al, instead of Qing Kong et al.

There are a few instances where there is no consistency in the writing style. For instance, the authors use GSDMB1/2 in one section, then in another use GSDMB1-2. Same as in GSDMB3-4 and GSDMB3/4.

The sections on asthma, inflammatory bowel disease, and even chronic sinusitis are less developed. There are tons of literature about GSDMB and these diseases including GWAS association and mechanisms. Because the authors included researchers and clinicians in molecular biology and inflammatory diseases as their intended audience (which is also reflected in the title of this literature review), this reviewer believes that these subsections should be improved.

If proptosis execution is a function of GSDMB, could the authors comment on whether this may be a target for prevention of diseases?

A figure showing the cellular functions of GSDMB will enhance the paper.

Reviewer 2 ·

Basic reporting

The paper summarizes the role of GSDMB in human diseases. The interest in gasdermin family has grown recently and new papers are needed to review available information about them. The Authors chose particularly GSDMB.
The introduction sufficiently provides a basic information about GSDMB, the structure and functions are thoroughly described.

Experimental design

The methods are quite thoroughly described.
The paper is organized in a logical manner, easy to read and understand.

Validity of the findings

Conclusions could be longer and more extensive, I would add information about potential therapeutic targets associated with GSDMB

Additional comments

Despite the references are prepared alphabetically, there are no consecutive numbers which makes it difficult to follow and separate one from the other.

line 293 - add diabetes MELLITUS

the names of genes should be in italics

I would perhaps add a little piece of information about GSDMB in different infectious diseases - there is literature to support that.

Reviewer 3 ·

Basic reporting

The manuscript explores the multifaceted role of Gasdermin B (GSDMB) in cancer and inflammatory diseases, with a focus on its molecular mechanisms and therapeutic potential. While the topic is highly relevant and comprehensive, there are issues related to grammar, clarity, citation adequacy, and organization that need to be addressed.

Detailed Comments
Abstract

Line 18: The sentence "Recent studies have revealed a range of functions for GSDMB..." could benefit from a brief mention of specific diseases or contexts for better clarity.
Line 23: Add a citation to support the claim that GSDMB facilitates pyroptosis.
Introduction

Line 38: Replace "Pyroptosis was defined in 2015 as..." with "Pyroptosis, first defined in 2015, is..."
Line 57: Rephrase "the GSDM gene family was first reported..." for clarity to "The GSDM gene family was first identified..."
Line 64: Expand the justification for this review by briefly comparing it with other recent reviews on the topic.
Characteristics of the GSDMB Gene and Protein Structure

Line 91: Provide additional context for the location of GSDMB on chromosome 17q21 by mentioning its association with specific diseases.
Line 99: Clarify the phrase "the molecular mechanisms by which these promoters regulate..." with specific examples or references.
Line 112: Add a citation supporting the statement on the genetic features like SNPs controlling tissue-specific expression of GSDMB.
Molecular Functions of the GSDMB Protein

Line 118: The phrase "binds to acidic phospholipids" should be clarified with examples or citations.
Line 132: Expand on the role of sulfatides in cancer progression and provide references for the claims.
Line 145: Rephrase "enhanced cell death caused by GSDMB..." for clarity to "the increase in cell death mediated by GSDMB..."
GSDMB and Cancer
Cite “Cancer statistics, 2024, 2024”. Then give intro in cancer therapy in general, cite NIH paper “Cancer treatments: Past, present, and future, 2024” (PMID: 38909530)for more information.
Line 164: Provide a citation supporting the claim that pyroptosis creates a tumor-promoting microenvironment in hypoxic regions.
Line 198: Add references for recent studies highlighting the nuclear localization of GSDMB and its transcriptional effects.
Line 217: Expand the explanation of why GSDMB co-expresses with ERBB2 in breast cancer.
discuss other cell death, especially recent discovered ferroptosis repoeted in (Targeting regulated cell death pathways in acute myeloid leukemia, 2023), other recently discovered cell death pathways should also be mentioned, such as cuproptosis (“Expression and potential immune involvement of cuproptosis in kidney renal clear cell carcinoma, 2023”,“Pan-cancer genetic analysis of cuproptosis and copper metabolism-related gene set, 2022”,“Pan-cancer profiles of the cuproptosis gene set, 2022”) and disulfidptosis (“Pan-cancer genetic analysis of disulfidptosis-related gene set, 2023”Actin cytoskeleton vulnerability to disulfide stress mediates disulfidptosis, 2023), discuss whether they affect the mechanisms in this study.
GSDMB and Genetic and (Auto)Inflammatory Diseases

Line 294: Clarify the link between GSDMB gene variants and type 1 diabetes with additional references.
Line 318: Add references for studies across different ethnic groups related to asthma susceptibility.
Advances in Tumor Therapy Involving GSDMB

Line 362: The phrase "GSDMB overexpression promotes tumorigenesis..." needs citations to support the claim.
Line 371: Elaborate on the mechanism by which NK cells secrete GZMA to induce pyroptosis.
Conclusions

Line 396: Expand on the "double-edged sword" nature of GSDMB with specific examples from the main text.

Experimental design

ok

Validity of the findings

ok

Additional comments

ok

Reviewer 4 ·

Basic reporting

1.The manuscript discusses GSDMB’s dual roles in cancer progression—its pro-tumor and anti-tumor effects—yet this aspect remains somewhat unclear. The manuscript does not provide a thorough explanation of the contexts in which GSDMB’s pyroptotic and non-pyroptotic functions impact cancer progression. Further elaboration on the mechanisms behind this duality, including specific examples of when GSDMB promotes or inhibits tumor growth, would enhance clarity.
2.While the review mentions GSDMB isoforms, it lacks in-depth discussion of the mechanistic differences between these isoforms. The functional distinctions between the isoforms, particularly in relation to cancer progression and treatment responses, need to be explored more thoroughly. The review could benefit from a more detailed analysis of how the various isoforms contribute to cancer and their potential as therapeutic targets.
3.In the section titled "GSDMB and Cancer," the manuscript could benefit from a deeper exploration of the complex relationship between GSDMB and the tumor microenvironment (TME). The TME has become a central focus of cancer research due to its critical role in tumor progression, immune modulation, and therapeutic resistance. Including a discussion on how GSDMB interacts with various components of the TME, such as immune cells, stromal cells, and extracellular matrix, would significantly enhance the manuscript's relevance and depth. Including references to recent studies on the role of GSDMB in the TME, such as those found in PMID: 39726592, PMID: 38962009, and PMID: 39267056, would strengthen the manuscript and provide additional credibility to the discussion.
4.The review briefly touches upon the controversial aspects of GSDMB’s role in tumor progression, especially regarding its pyroptotic functions. However, this discussion could be expanded to include a more detailed comparison of conflicting viewpoints and data in the literature. A deeper exploration of these controversies would provide a more balanced perspective and help identify gaps in current research.
5.Some sections require proofreading to correct minor typographical errors.

Experimental design

no comment

Validity of the findings

no comment

Additional comments

no comment

---

## Round 0.2 · Minor Revisions

As you make the final adjustments to your manuscript, please take a moment to carefully review and address the minor revisions suggested by the two reviewers.

Reviewer 1 ·

Basic reporting

OK

Experimental design

OK

Validity of the findings

N/A

Additional comments

The authors have addressed my comments.

Minor comment: With regards to writing style, there are instances where the authors use the full name of first author when describing their work in the main context, yet in other instances only last name was used. For example, in Line 134"Sara S. Oltra et al." was used while in Line 193, "Chen et al..." was used. I suggest to be consistent throughout the manuscript.

Reviewer 3 ·

Basic reporting

good

Experimental design

good

Validity of the findings

good

Additional comments

Line 651 for future study. Recent studies have highlighted advancements in liquid biopsies for cancer diagnostics and monitoring. Research such as “Updates on liquid biopsies in neuroblastoma for treatment response, relapse and recurrence assessment, 2024”demonstrates the utility of circulating tumor DNA (ctDNA) detection through liquid biopsy techniques. Additionally, emerging sequencing technologies have improved the sensitivity and specificity of DNA analysis, such as “Development of a molecular barcode detection system for pancreaticobiliary malignancies and comparison with next-generation sequencing, 2024”. Also the methylation is also used for detection, reported in “Methylation signatures as biomarkers for non-invasive early detection of breast cancer: A systematic review of the literature, 2024”. Please cited these related papers and discuss: consider whether the mechanisms discussed in this study could be identified through these diagnosis methods.

In line 397when you mentioned TCGA, you should also discuss the bias from TCGA, refer to “Genetic expression in cancer research: Challenges and complexity, 2024” and “Technical and Biological Biases in Bulk Transcriptomic Data Mining for Cancer Research, 2025”. Because this impact the conclusion of this study.

Reviewer 4 ·

Basic reporting

good

Experimental design

good

Validity of the findings

good

Additional comments

good

---

## Round 0.3 · accepted · Accept

I am pleased to see that all the minor comments have been addressed, and I am happy to accept your manuscript. Excellent work!